# Comparative transcriptome analysis of the cold resistance of the sterile rice line 33S

Hongjun Xie[1◉], Mingdong Zhu[1◉], Yaying Yu[1], Xiaoshan Zeng[1], Guohua Tang[1], Yonghong Duan[1], Jianlong Wang[2]*, Yinghong Yu[3]*

1 Hunan Rice Research Institute, Key Laboratory of Indica Rice Genetics and Breeding in the Middle and Lower Reaches of Yangtze River Valley, Changsha, China, 2 Southern Regional Collaborative Innovation Center for Grain and Oil Crops, College of Agronomy, Hunan Agricultural University, Changsha, China, 3 Hunan Academy of Agricultural Sciences, Changsha, China

◉ These authors contributed equally to this work.
* wjl9678@126.com (JW); yyh30678@163.com (YY)

**Data Availability Statement:** All relevant data are within the paper and its Supporting information files.

**Funding:** This work was funded by the National Key R&D Program of China (2018YFD0301001).

## Abstract

Rice (*Oryza sativa* L.) is one of the most important species for food production worldwide. Low temperature is a major abiotic factor that affects rice germination and reproduction. Here, the underlying regulatory mechanism in seedlings of a TGMS variety (33S) and a cold-sensitive variety (Nipponbare) was investigated by comparative transcriptome. There were 795 differentially expressed genes (DEGs) identified only in cold-treated 33S, suggesting that 33S had a unique cold-resistance system. Functional and enrichment analysis of these DEGs revealed that, in 33S, several metabolic pathways, such as photosynthesis, amino acid metabolism, secondary metabolite biosynthesis, were significantly repressed. Moreover, pathways related to growth and development, including starch and sucrose metabolism, and DNA biosynthesis and damage response/repair, were significantly enhanced. The expression of genes related to nutrient reserve activity were significantly up-regulated in 33S. Finally, three NAC and several ERF transcription factors were predicted to be important in this transcriptional reprogramming. This present work provides valuable information for future investigations of low-temperature response mechanisms and genetic improvement of cold-tolerant rice seedlings.

## Introduction

Rice (*Oryza sativa* L.) is one of the most important staple crop species, feeding more than half of the global population. Increasing rice yield is a priority to ensure global food security. Low temperature is one of the major environmental stresses that negatively impacts plant growth and yield potential. Due to its tropical and subtropical origin, rice is sensitive to low temperature [1]. The optimal temperature is 25–35°C for rice at the germination stage. Temperatures below 15°C would lead to a number of developmental damage in rice, including reduced germination rate, delayed seedling emergence and initial seedling growth, and high seedling mortality [2]. In the reproductive phase, low temperature can cause sterility by interrupting meiosis and mitosis, thus resulting in the failed formation of mature microspores [3]. The current high-yield production of superhybrid rice cultivars are frequently affected by cold stress in tropical or subtropical areas. Thus, improving rice cold stress tolerance could help maintain

The funders had no role in study design, data collection and analysis, decision to publish, or preparation of the manuscript.

**Competing interests:** The authors have declared that no competing interests exist.

rice production in regions where it is currently grown and expand production into northern areas with lower annual temperatures [4].

Hybrid rice cultivars have a yield advantage that 10–20% greater than that of conventional inbred cultivars [5]. In the past few decades, numerous hybrid rice cultivars have been developed in more than 40 countries and have played a critical role in the global food supply [6]. Currently, F1 Hybrid rice is mainly produced using two systems: the cytoplasmic male sterility-based three-line system and the thermo/photoperiod-sensitive genic male sterile-based two-line system. The first is the traditional three-line system, which requires a cytoplasmic male-sterile (CMS) line, a restorer line, and a maintainer line to produce hybrid seeds and to maintain the CMS line [7, 8]. The other is a two-line breeding system, which uses a thermo-/photoperiod-sensitive genic male-sterile (T/PGMS) line as both a sterility line and a maintainer line under specific environmental conditions [9]. Compared with the three-line system, the two-line hybrid rice system has increased the yield potential of rice by 10% [10]. In addition, the two-line system is more cost-effective, produces higher quality grain, is more flexible in terms of germplasm, and is simpler to operate [11]. Previously, we developed a cold-resistant thermosensitive genic male-sterile (TGMS) line, 33S, with a critical temperature of <23˚C [12]. This line has been widely used in rice breeding in China. With the use of 33S as the sterile line, 38 new hybrid rice varieties have been approved, and the total increased production area is more than one million hectares. We found that 33S seedlings had strong low-temperature tolerance. In the present study, the expression profile of the thermosensitive genie male-sterile variety S33 was analyzed by transcriptome sequencing to understand the molecular mechanism of low-temperature adaptation in T/PGMS rice lines.

## Materials and methods

### Plant materials and stress treatment

The japonica rice (*O. sativa* L.) variety Nipponbare (a cold-sensitive material) and the indica TGMS variety 33S (a cold-resistant material) were used for transcriptomic analysis. Rice seeds were germinated at 30˚C under a photoperiod of 12 h of light/12 h of darkness. For the low-temperature treatment, fifty one-week-old seedlings were subjected to 15˚C for seven days under a photoperiod of 12 h of light/12 h of darkness. After the treatment, leaf tissues randomly collected from ten seedlings were immediately frozen in liquid nitrogen and then stored at -80˚C until RNA extraction. The remaining seedlings were moved to normal conditions and allowed to grow for another week.

### RNA extraction and library construction

Total RNA was extracted from three independent biological replicates using the ethanol precipitation protocol and CTAB-pBIOZOL reagent according to the manufacturer's instructions. Total RNA was qualified and quantified using a NanoDrop instrument (Thermo, USA) and an Agilent 2100 Bioanalyzer (Thermo Fisher Scientific, MA, USA). A total amount of 3 µg of RNA per sample was used as input material for the RNA sample preparations. Sequencing libraries were generated using a NEBNext® Ultra™ RNA Library Prep Kit for Illumina® (NEB, USA) following the manufacturer's recommendations. The library preparations were sequenced on an Illumina HiSeq 2500 platform in accordance with a 100 bp paired-end pattern (PE-100). Clean data were obtained by removing reads containing adapter, reads containing ploy-N and low quality reads from raw data, and all transcriptome analyses were based on the clean data.

## RNA sequencing (RNA-seq) data analysis

Reference genome and gene model annotation files were downloaded from Rice Genome Annotation Project (GCF_000005425.2_Build_4.0_genomic) [13]. The index of the reference genome was built using Bowtie v2.0.6 [14], and paired-end clean reads were aligned to the reference genome using TopHat v2.0.9 [15]. To quantify the gene expression levels, HTSeq v0.6.1 was used to count the read numbers mapped to each gene [16]. Afterward, the reads per kilobase of exon model per million mapped reads (RPKM) of each gene was calculated based on the length of the gene and read count mapped to that gene. Differentially expressed genes (DEGs) were identified using the DESeq R package (version 1.10.1) between different samples with parameters: adjusted p value$< 0.05$ and $|log2FC| > = 1$ [17]. The resulting P-values were adjusted using the Benjamini and Hochberg's approach for controlling the false discovery rate.

## Availability of data and materials

The raw reads produced in this study were deposited in the NCBI SRA with the submission number SUB10060963 under bio Project PRJNA749812.

## Functional annotation of DEGs

For Gene Ontology (GO) enrichment analysis, the Blast2GO software (version 2.3.5) was used to identify the DEGs according to biological process, molecular function and cellular component ontologies with a threshold E value $\leq 10^{-5}$ [18]. KOBAS software was used for the pathway enrichment and annotation analysis of the DEGs by comparison to the Kyoto Encyclopedia of Genes and Genomes (KEGG) database [19].

## RT-qPCR validation

The RNA-seq data were validated via RT-qPCR analyses using an Applied Biosystems StepOnePlus™ Real-Time System with SYBR® Select Master Mix (2X) (ABI, USA). The reaction protocol was as follows: 95˚C for 3 min; 40 cycles at 95˚C for 15 s, 55˚C for 15 s, and 72˚C for 20 s; and then 72˚C for 5 min. All the specific primers used were presented in S1 Table. Three technical replicates were included per sample. The rice *actin 1* gene (gene locus: Os03g0718100) was used as an internal standard [20]. The relative expression value of the different genes were calculated using the $2^{-\Delta\Delta Ct}$ method.

# Results and discussion

## 33S is more cold tolerant than Nipponbare

33S is an indica TGMS line that was selected for a 15-year hybrid from Peiai 64S, Yue 4B and Ganwanxian 30. Seedlings of the low-temperature-tolerant rice variety 33S and the low-temperature-sensitive variety Nipponbare were sampled after 7 days of low-temperature treatment at 15˚C (Fig 1). After treatment, 12.4±0.08% of the Nipponbare seedlings died, while no visible growth effects were observed of the 33S seedlings after low temperature treatment. Additionally, the length of the shoots of Nipponbare was obviously reduced, while the length of the shoots and roots of 33S showed no differences (Fig 1).

## Identification of DEGs and functional classification

In order to elucidate the cold response mechanism, seedlings of the low-temperature-tolerant rice variety 33S and the low-temperature-sensitive variety Nipponbare were sampled after 7

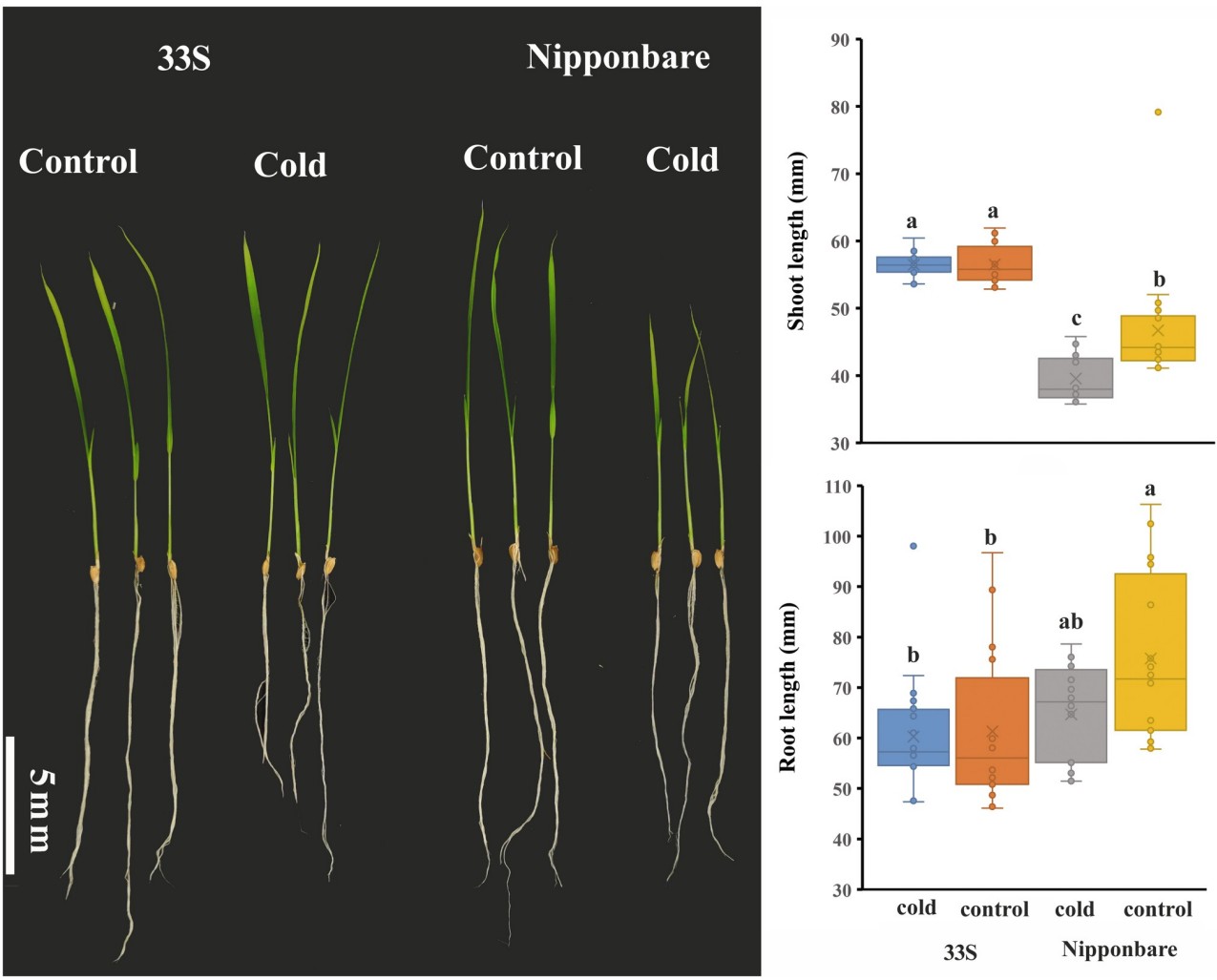

**Fig 1. Phenotype of 33S and Nipponbare after exposed to cold treatment. A**. Seedling of 33S and Nipponbare. **B**. Shoot length of 33S and Nipponbare seedlings after cold treatment. **C**. Root length of 33S and Nipponbare seedlings after cold treatment. Different letters above the box indicate a significant difference at P < 0.05. Error bar was statistic by standard deviation.

days of low-temperature treatment at 15˚C. The samples were subjected to total RNA extraction and RNA-seq analysis using the Illumina HiSeq 2500 platform.

The genes and RPKM values of six samples were calculated. Differential expression analysis under two conditions was performed using the DESeq R package v1.10.1 using a threshold of P < 0.05 and |log2(fold change)| > = 1. A total of 938 DEGs were detected in 33S after cold treatment, including 347 upregulated genes and 591 downregulated genes (S2 and S3 Tables) (Fig 2). In Nipponbare, a total of 762 DEGs were detected after cold treatment, of which 398 were upregulated (S4 Table) and 364 were downregulated (Fig 2, S5 Table). Compared with the Nipponbare variety, more genes were down-regulated in 33S variety after seven days low temperature treatment. In contrast, more up-regulated genes were identified in Nipponbare variety. Venn diagram was constructed to analyze the common DEGs between 33S and Nipponbare. The result showed that only 143 common DEGs were detected between the two varieties (Fig 2C, S6 Table).

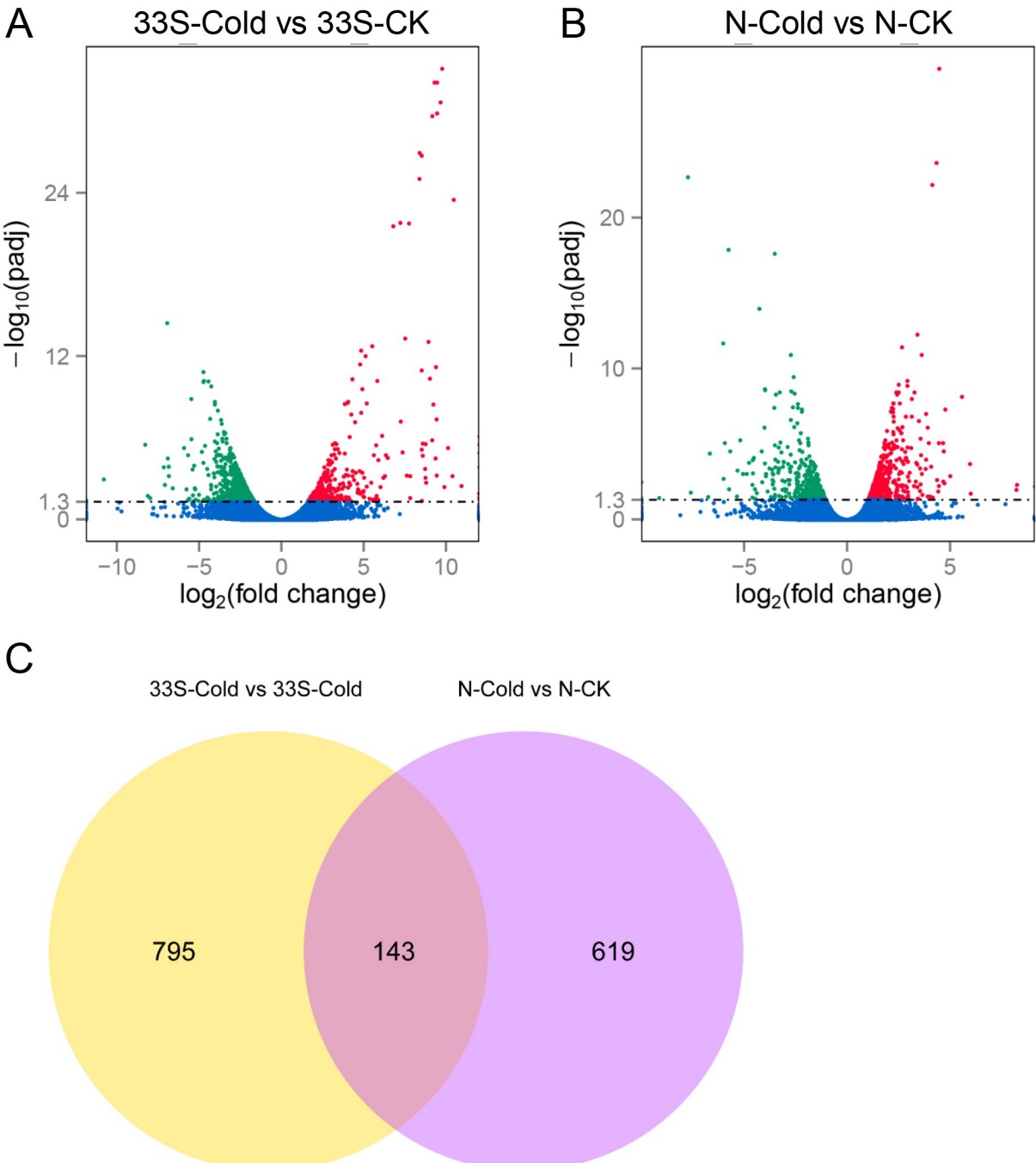

**Fig 2. Number of DEGs after low temperature treatment in 33S and Nipponbare. A**. Significantly up- or down-regulated genes in 33S-Cold vs. 33S-CK. **B**. Significantly up- or down-regulated genes in N-Cold vs. N-CK. **C**. Venn diagram indicating the number of DEGs common to 33S and Nipponbare. N indicates the variety of Nipponbare, CK indicates the control seedling not treated by cold temperature. DEGs were screened using a threshold of P <0.05 and |log2FoldChange| > = 1.

To further characterize genes affected by low temperature stress, all DEGs were subjected to Gene Ontology (GO) enrichment analysis. According to their functions, the DEGs were classified into three classes, including biological process, cellular component and molecular function (S7 and S8 Tables). The top 30 enrichment terms for 33S and Nipponbare were listed

in Fig 3A and 3B, respectively. Possibly due to the difference in low temperature tolerance, the top 30 GO enrichment terms were dramatically different in 33S and Nipponbare varieties, even though few terms were enriched in cellular component for both varieties (Fig 3A and 3B). For 33S variety, within the top 30 most enrichment terms, there were ten, five, and fifteen enriched terms belonging to the biological process, cellular component, and molecular

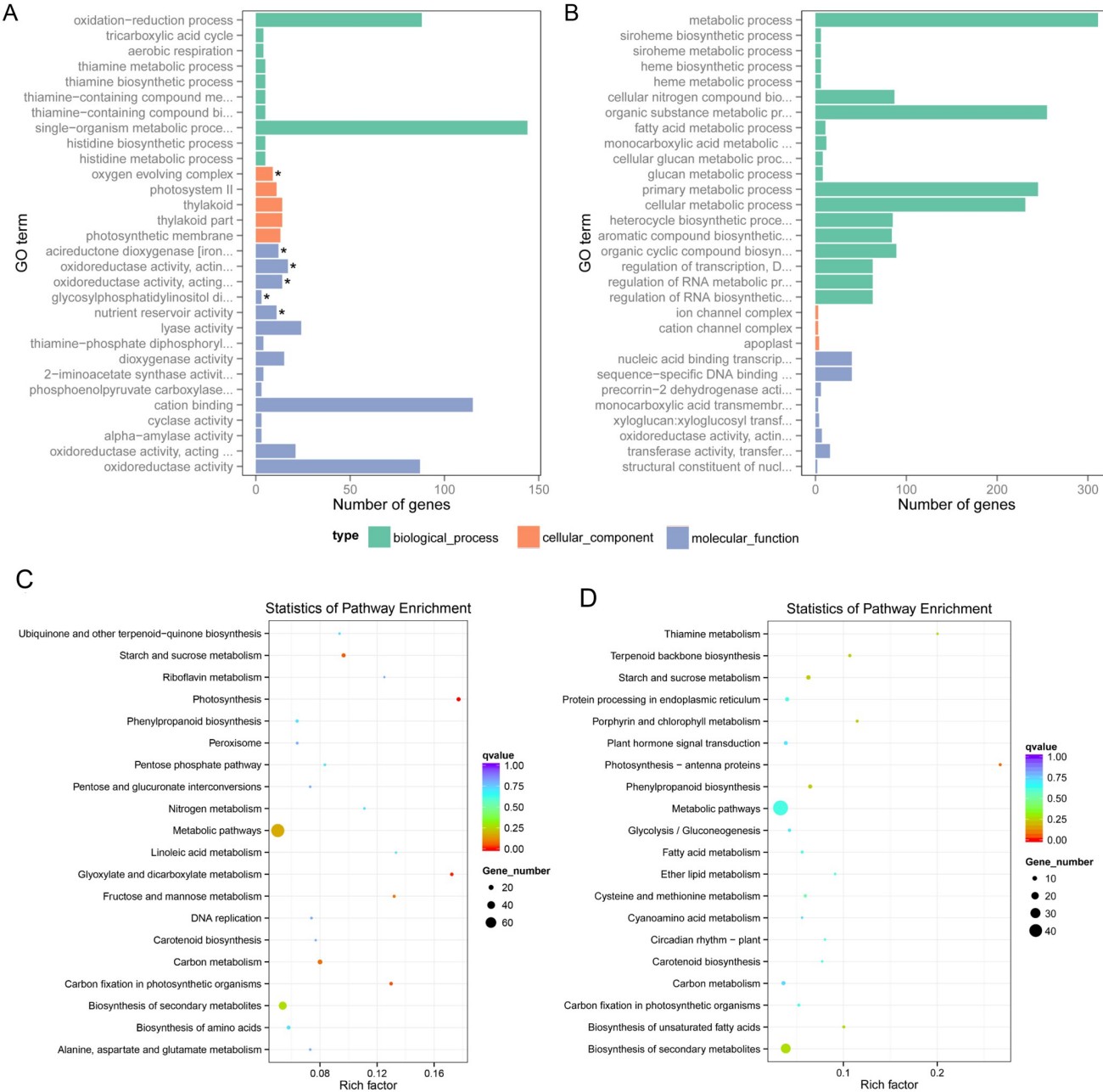

**Fig 3. Functional analysis of cold responsive genes. A**. GO enrichment analysis of the DEGs in 33S rice seedlings after seven days of low-temperature treatment. **B**. GO enrichment analysis of the DEGs in Nipponbare rice seedlings after seven days of low-temperature treatment. The DEGs were classified into three main GO categories (biological processes, molecular functions and cellular components). **C**. KEGG pathway enrichment analysis of DEGs in 33S rice seedlings after seven days of low-temperature treatment. **D**. KEGG pathway enrichment analysis of DEGs in Nipponbare rice seedlings after seven days of low-temperature treatment. The P-value to determine the enrichment significance was calculated through hypergeometric distribution.

function categories, respectively. However, for Nipponbare variety, the number were nineteen, three, and eight. In the biological process category, a high percentage of DEGs were associated with oxidation-reduction process and single-organism metabolic process for 33S variety. In contrast, a high percentage of DEGs were enriched in metabolic process, cellular nitrogen compound biosynthesis, primary metabolic process, cellular metabolic process, heterocycle biosynthetic process, aromatic compound biosynthetic process, organic cyclic compound biosynthesis, regulation of transcription, regulation of RNA metabolic process, and regulation of RNA biosynthetic process. A relatively small number of DEGs were classified into terms of cellular component category for both varieties. In the molecular function category, a high percentage of DEGs were classified into oxidoreductase activity and cation binding terms. However, a relatively small number DGEs were fall into the molecular function category in Nipponbare variety. Taken together, all these results suggested that these two rice varieties responses differently under low temperature condition.

The low-temperature-responsive DEGs were further mapped to terms in the KEGG database. In 33S, 349 DEGs were significantly enriched in 67 metabolic pathways or signal transduction pathways, while in Nipponbare, 226 DEGs were enriched in 61 metabolic pathways or signal transduction pathways (S9 and S10 Tables). Seven pathways out of the top 20 significantly enriched pathways were significantly enriched in the 33S and Nipponbare varieties, including starch and sucrose metabolism, phenylpropanoid biosynthesis, metabolic pathways, carotenoid biosynthesis, carbon metabolism, carbon fixation in photosynthetic organisms, and biosynthesis of secondary metabolites (Fig 3C and 3D). In addition, the DEGs were also significantly enriched in pathways such as plant hormone signal transduction, phenylpropanoid biosynthesis, protein processing in the endoplasmic reticulum (ER), and photosynthesis pathways in both varieties. Metabolic pathways were mostly affected, followed by biosynthesis of secondary metabolites, during low-temperature treatment in both varieties. The KEGG enrichment analysis provided valuable information for investigating the pathways and gene functions involved in the cold stress response.

## Validation of transcriptome data by RT-qPCR analyses

To validate the results of the RNA-seq data, RNA samples extracted for RNA-seq were also subjected to qRT-PCR analysis. A total of eight DEGs were randomly selected for quantitative qRT-PCR analysis, including four down-regulated genes (*Os01g0124000*, *Os01g0971800*, *Os05g0382600*, and *Os06g0474800*) and four up-regulated genes (*Os03g0161900*, *Os03g0293000*, *Os02g0685200*, and *Os10g0509700*). As anticipated, the expression profiles of these genes according to the qRT-PCR results were essentially consistent with those generated from RNA-seq, suggesting that the DEGs resulting from RNA-seq were credible for further analysis (Fig 4).

## Low-temperature-responsive genes involved in hormone signal transduction

Cytokinin plays important roles not only in various plant growth and development processes but also in abiotic and biotic stress responses [21, 22]. Two-component A-type response regulators (ARRs) are part of the cytokinin signaling cascade, and 15 ARRs were identified in the rice genome [23]. In this study, four ARR genes (*Os02g0557800*, *Os11g0143300*, *Os12g0139400*, and *Os04g0673300*, encoding ARR3, ARR6, ARR9, and ARR10, respectively) responded to low-temperature treatment in the 33S variety. Three ARR genes (*Os02g0557800*, *Os12g0139400*, and *Os04g0673300*, encoding ARR3, ARR9, and ARR10, respectively) were responsive to low-temperature treatment in the Nipponbare variety (Table 1). ARR9 expression was previously reported to be regulated by the circadian clock in *Arabidopsis* [24]. In rice, ARR9 expression

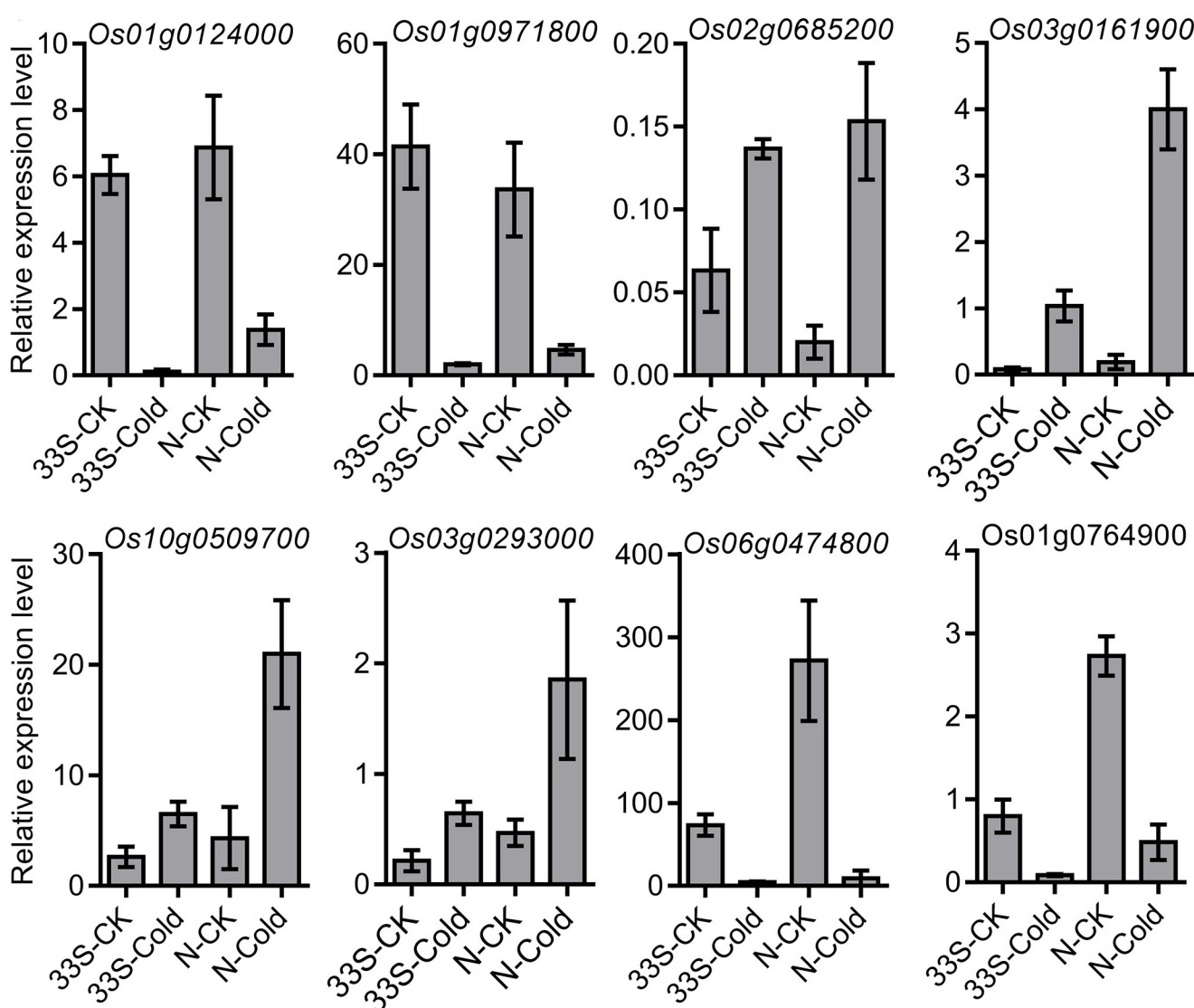

**Fig 4. RT-qPCR verification of expression profiles obtained through RNA-seq.** Data are mean ± SD of three biological replicates. Asterisks indicate significant difference compared to control samples ($^*P < 0.05$; $^{**}P < 0.01$). The *actin 1* gene was used as an internal standard gene was used as an internal reference.

was shown to be downregulated under cold stress in three cold-tolerant genotypes [25]. Furthermore, rice ARR9 and ARR10 genes were strongly positively correlated with the zinc finger transcription factor (TF) Drought and Salt Tolerance (DST) [26]. The loss of DST function increases stomatal closure and reduces stomatal density, consequently resulting in enhanced drought and salt tolerance in rice. Our results suggested that the ARR-mediated cytokinin signaling cascade might be differentially involved in the low-temperature response of the two varieties.

The C-repeat-binding factor (CBF)-dependent cold signaling pathway plays an important regulatory role in the low-temperature-responsive signaling pathway [27]. OST1 (*OPEN STOMATA 1*) acts upstream of CBFs to positively regulate freezing tolerance in *Arabidopsis* and increases the freezing resistance of plants by phosphorylating and stabilizing the key transcription factor ICE1, which acts upstream of the CBF genes [28]. In addition, OST1 can indirectly regulate the stability

**Table 1. List of DEGs enriched in plant hormone signal transduction.**

| Variety | Loci | Pathway | log2FoldChange |
|---------|------|---------|----------------|
| 33S | *Os04g0673300* | Two-component response regulator ARR6 | 1.7729 |
| | *Os11g0143300* | Two-component response regulator ARR9 | 2.1111 |
| | *Os12g0139400* | Two-component response regulator ARR10 | 2.0053 |
| | *Os02g0557800* | Two-component response regulator ARR3 | 1.8845 |
| | *Os05g0186100* | Histidine-containing phosphotransfer protein 4 | 3.0962 |
| | *Os03g0610900* | Serine/threonine-protein kinase SAPK10 | −1.9516 |
| | *Os12g0586100* | Serine/threonine-protein kinase SAPK9 | −2.0314 |
| | *Os06g0527800* | Abscisic acid receptor PYL8 | 5.8919 |
| Nipponbare | *Os01g0221100* | Probable indole-3-acetic acid-amido synthetase GH3.3 | −1.9541 |
| | *Os01g0583100* | Probable protein phosphatase 2C 6 | 1.3689 |
| | *Os11g0143300* | Two-component response regulator ARR9 | 2.2694 |
| | *Os12g0139400* | Two-component response regulator ARR10 | 2.0143 |
| | *Os02g0557800* | Two-component response regulator ARR3 | 1.5918 |
| | *Os05g0186100* | Histidine-containing phosphotransfer protein 4 | 2.5623 |
| | *Os05g0457200* | Probable protein phosphatase 2C 49 | 3.0365 |

of CBF proteins and enhance the freezing resistance of plants by phosphorylating the new poly-peptide chain-coupled protein complex β subunit BTF3 [29]. There are three OST1 homologs in the rice genome (*Os03g0610900*, *Os12g0586100* and *Os03g0764800*). A previous study demonstrated that the expression of *Os03g0610900* in rice was influenced by the methylation level of its promoter region, and low-temperature treatment could decrease the methylation level [30]. We found that two of the homologous genes (*Os03g0610900* and *Os12g0586100*) responded to low-temperature treatment only in the 33S variety (Table 1). Additionally, no Ser/Thr protein kinase-encoding genes were identified in Nipponbare, which might partly explain the difference in the low-temperature tolerance between these two varieties. The different response patterns of *Os03g0610900* and *Os12g0586100* to low temperature need further investigation. The OST1 signaling pathway has not been clearly elucidated in rice, a monocotyledonous species, and the functions of some of the homologous genes in this pathway have not yet been identified. Therefore, it is necessary to determine the physiological and biochemical functions of these genes in rice in further studies.

PYL8 was previously reported to play an important role for ABA signaling and drought stress responses, and overexpression of *PLY8* in *Arabidopsis* could increase the ABA sensitivity and drought tolerance [31]. According to our results, the expression of the *PYL8* gene was dramatically induced in response to low-temperature treatment in 33S, which indicated that PYL8 might regulate plant cold tolerance through ABA signaling pathway (Table 1). However, the expression of this gene was not induced in the low temperature-sensitive Nipponbare variety.

## Low-temperature-responsive genes involved in protein processing in the ER

Biotic and abiotic stress could result in protein misfolding and the accumulation of unfolded proteins. The ER responds to unfolded proteins, which can cause deleterious effects, in its lumen (ER stress) by activating intracellular signal transduction pathways; this process is collectively termed the unfolded protein response (UPR) [32]. The ubiquitin-mediated protein degradation pathway, which has been demonstrated as a key regulatory mechanism in response to biotic stress, is responsible for the major portion of specific cellular misfolded

protein degradation [33]. In this study, five DEGs were enriched in protein processing in the ER pathway after 7 days of low-temperature treatment in 33S. Of the five DEGs, three (*Os01g0369200*, *Os02g0639800*, and *Os04g0667800*) were predicted to participate in the ubiquitin-mediated protein degradation pathway (Table 2). Another two DEGs (*Os01g0135900* and *Os07g0517100*) were annotated as heat-shock proteins (HSPs), which also play an important role in abiotic stress. As molecular chaperones, HSPs are responsible for protein folding, assembly, translocation and degradation of damaged proteins and play critical roles in protecting plants against stress by stabilizing proteins and membranes [34]. In contrast, eight DEGs were enriched in protein processing in the ER pathway in Nipponbare, which suggested that the cold-sensitive variety Nipponbare experienced more severe ER stress after low-temperature treatment (Table 2). In addition, the enriched terms/pathways of DEGs between these two varieties were totally different, indicating that they responded differently to low-temperature treatment.

## Low temperature-responsive transcription factors

Transcription factors play a critical role in biotic and abiotic responses in plants. In this present paper, a total of 40 and 42 TFs were identified in the DEGs from 33S and Nipponbare varieties after 7 days of low temperature treatment, respectively (Table 3). The expression of TF families showed strong responses to low temperature treatment included AP2/ERF-ERF, B3, bHLH, bZIP, C2C2-CO-like, C2H2, C3H, B-box zinc finger protein, GARP-G2-like, GRAS, HB-BELL, HB-HD-ZIP, HMG, HSF, LOB, MADS-MIKC, MYB, MYB-related, NAC, SBP, NF-YC, zf-HD. Most of these TF families had been reported to play an important role in abiotic stress including cold stress [35–37] and some of these TFs had bene utilized to improve plant abiotic stress tolerance by genetic transformation technology [38–40].

However, the numbers and members of differently expressed TFs showed a dramatic difference between 33S and Nipponbare variety. For example, five C2H2 (*Os01g0871200*, *Os02g0709000*, *Os03g0239300*, *Os06g0166200*, and *Os09g0431900*) family members were identified in 33S, however, no gene belonged to this family was detected in Nipponbare (Table 3, S11 and S12 Tables). In addition, four NAC (*Os01g0393100*, *Os02g0214500*, *Os04g0460900*, and *Os11g0512000*) family members responded to low temperature treatment, but just one

**Table 2. List of DEGs enriched in protein processing in endoplasmic reticulum pathway.**

| Variety | Loci | Description | log2FoldChange |
|---|---|---|---|
| 33S | *Os01g0369200* | Cullin-1 | 2.2995 |
| | *Os02g0639800* | E3 ubiquitin-protein ligase RMA3 | −1.8144 |
| | *Os01g0135900* | 17.9 kDa heat shock protein 2 | −1.7106 |
| | *Os04g0667800* | Ubiquitin-conjugating enzyme E2 | −1.8764 |
| | *Os07g0517100* | 18.8 kDa class V heat shock protein | −2.7372 |
| Nipponbare | *Os06g0219500* | 26.2 kDa heat shock protein | 1.8964 |
| | *Os01g0895600* | Calreticulin-3 | −1.1534 |
| | *Os02g0758000* | 24.1 kDa heat shock protein | 3.5681 |
| | *Os01g0840100* | Heat shock cognate 70 kDa protein | 2.6097 |
| | *Os03g0832200* | Calreticulin | −2.2437 |
| | *Os01g0184100* | 18.0 kDa class II heat shock protein | 3.3999 |
| | *Os03g0799900* | SEC12-like protein 1 | −1.7853 |
| | *Os11g0703900* | Heat shock cognate 70 kDa protein 2 | −1.1669 |

**Table 3. Transcription factors response to low temperature treatment in 33S and Nipponbare.**

| TF family | Number of genes identified | |
| --- | --- | --- |
| | 33S | Nipponbare |
| AP2/ERF | 2 | 5 |
| B3 | 1 | 1 |
| bHLH | 3 | 0 |
| bZIP | 3 | 2 |
| C2C2-CO-like | 2 | 4 |
| C2C2-Dof | 0 | 1 |
| C2H2 | 5 | 0 |
| C3H | 0 | 1 |
| B-box zinc finger protein | 1 | 2 |
| GARP-G2-like | 2 | 1 |
| GNAT | 2 | 1 |
| GRAS | 1 | 2 |
| HB-BELL | 1 | 0 |
| HB-HD-ZIP | 2 | 3 |
| HMG | 1 | 1 |
| HSF | 3 | 5 |
| LOB | 1 | 0 |
| MADS-MIKC | 1 | 3 |
| MYB | 1 | 3 |
| MYB-related | 3 | 4 |
| NAC | 4 | 1 |
| SBP | 0 | 1 |
| NF-YC | 1 | 0 |
| zf-HD | 1 | 1 |

NAC (*Os03g0815100*) gene was identified in Nipponbare (Table 3, S11 and S12 Tables). These differences were possibly due to the different response mechanism to low temperature stress.

bHLH (basic Helix-Loop-Helix) proteins are the second largest transcription factor families in plants and plays an important role in both plant development and abiotic stress responses [41]. bHLH genes had been demonstrated to be involved in cold response in a number of plants and overexpression of bHLH genes could enhance the cold tolerance in plants like apple, rice, and pummelo [42–45]. In particularly, *OsbHLH1* gene from rice was demonstrated to be involved in cold stress response [44]. Genetic transformation of rice further demonstrated that overexpression of *OsbHLH1* in rice could increase the cold tolerance during the germination and seedling stages [46]. In addition, most recently report showed that OsbHLH002 protein from rice was involved in cold response downstream of the phosphatase protein OsPP2C27 [47]. In our study, the expression of three bHLH genes (*Os01g0566800*, *Os01g0577300*, and *Os01g0952800*) were repressed by low temperature, which indicated that these genes might negatively correlated with the low temperature tolerance.

C2H2 type zinc finger proteins participate in responses to different environmental stresses in plant, including cold and drought stress [48]. In banana fruit, *MaC2H2-1*, *MaC2H2-2* and *MaC2H2-3* were cold inducible in the peel during low temperature storage, and MaC2H2s were proposed to be involved in cold stress response via repressing the transcription of MaICE1 [49]. Overexpression of a soybean C2H2-type zinc finger gene *GmZF1* could enhance the cold tolerance of transgenic *Arabidopsis* [50]. In this present study, the expression of five

C2H2-type zinc finger TFs showed response to low temperature treatment, one of which was induced (*Os02g0709000*) and the other four were repressed (*Os01g0871200*, *Os03g0239300*, *Os06g0166200*, and *Os09g0431900*). However, the function of these genes in rice and their homologous in other plant species have been characterized.

The NAC (NAM, ATAF and CUC) proteins constitute a large transcription factor family with more than 150 members in rice and a number of them have been demonstrated to play crucial roles in plant abiotic stress response [51]. The SNAC2 gene from rice could be induced by abiotic stress, including drought, salinity, cold, and wounding. The transgenic experiment results showed that overexpression of SNAC2 could significantly enhance the severe cold stress tolerance (4–8 ˚C for 5 days) of the transgenic plants [52]. In this present paper, four NAC genes (*Os01g0393100*, *Os02g0214500*, *Os04g0460900*, and *Os11g0512000*) showed response to low temperature treatment in 33S. The expression of one of the four NAC genes (*Os04g0460900*) downregulated by cold treatment, while the other three genes *Os01g0393100*, *Os02g0214500*, and *Os11g0512000*) were induced by low temperature stress (Table 2). *Os02g0214500* encodes the OsNAC20 protein and *Os01g0393100* encodes the OsNAC26 protein. Previous report demonstrated that *Os01g0393100* showed response to Combined abiotic stress, which indicated that *Os01g0393100* might play an important role in rice abiotic stress regulation [53]. Most recently, Os*NAC20* and *OsNAC26* genes were demonstrated to be expressed specifically in rice endosperm and regulated starch and storage protein synthesis [54]. In this paper, we found that the direct target genes of OsNAC20 and OsNAC26, including *Os02g0242600*, *Os02g0248800*, *Os02g0249000*, *Os02g0249600*, *Os02g0249800*, *Os02g0249900*, *Os02g0268100*, *Os05g0328333*, *Os05g0329100*, *Os05g0329400*, *Os05g0330600*, *Os05g0331550*, *Os05g0331800*, and *Os05g0332000*, were dramatically induced by low temperature in 33S variety (S2 Table). These genes encoding the endosperm specific proteins and starch biosynthesis genes. The decreased photosynthesis pathway genes and increased storage protein coding genes indicated that the low temperature tolerance 33S rice variety showed a dormant-like state to response the low temperature.

## 33S recruits specific DEGs to enhance its tolerance to cold stress

As indicated in Fig 2, 795 DEGs were detected only in 33S and not in Nipponbare (Fig 1C; S6 Table). These 795 DEGs suggested that 33S had a special adaptation system for cold stress (Fig 5). First, in 33S, basic nutrient and energy metabolism process were highly altered: the expression of genes involved in "photosynthesis" (the expression of 24 DEGs was downregulated in cold-treated 33S) and "amino acid metabolism" (8 downregulated DEGs of 9 DEGs) was almost entirely repressed, while that of "starch and sucrose metabolism" (11 upregulated DEGs of 14 DEGs) and "major CHO metabolism" (7 upregulated DEGs of 8 DEGs) completely increased. To reduce consumption, the expression of genes involved in several metabolic pathways, such as "secondary metabolite biosynthesis" (9 downregulated DEGs of 11 DEGs) and "redox" (10 downregulated DEGs of 11 DEGs), and genes encoding miscellaneous enzymes, including kinases (31 downregulated DEGs of 40 DEGs), was also predominantly repressed. Moreover, several pathways related to maintaining genetic stability, growth and development, such as "transport (mainly protein and phosphate transporters, or ABCs)" (22 upregulated DEGs of 36 DEGs) and "DNA biosynthesis and damage response/repair" (10 upregulated DEGs of 12 DEGs), were increased significantly. Furthermore, the expression of genes related to nutrient reserve activity was mostly upregulated, including 13 upregulated prolamin PPROL genes, whose expression increased 54.7-fold, and 11 upregulated glutelin type A and B genes, whose expression increased at least 166.6-fold. Overall, 33S was able to reduce unnecessary energy consumption by silencing most bioactivities and increasing

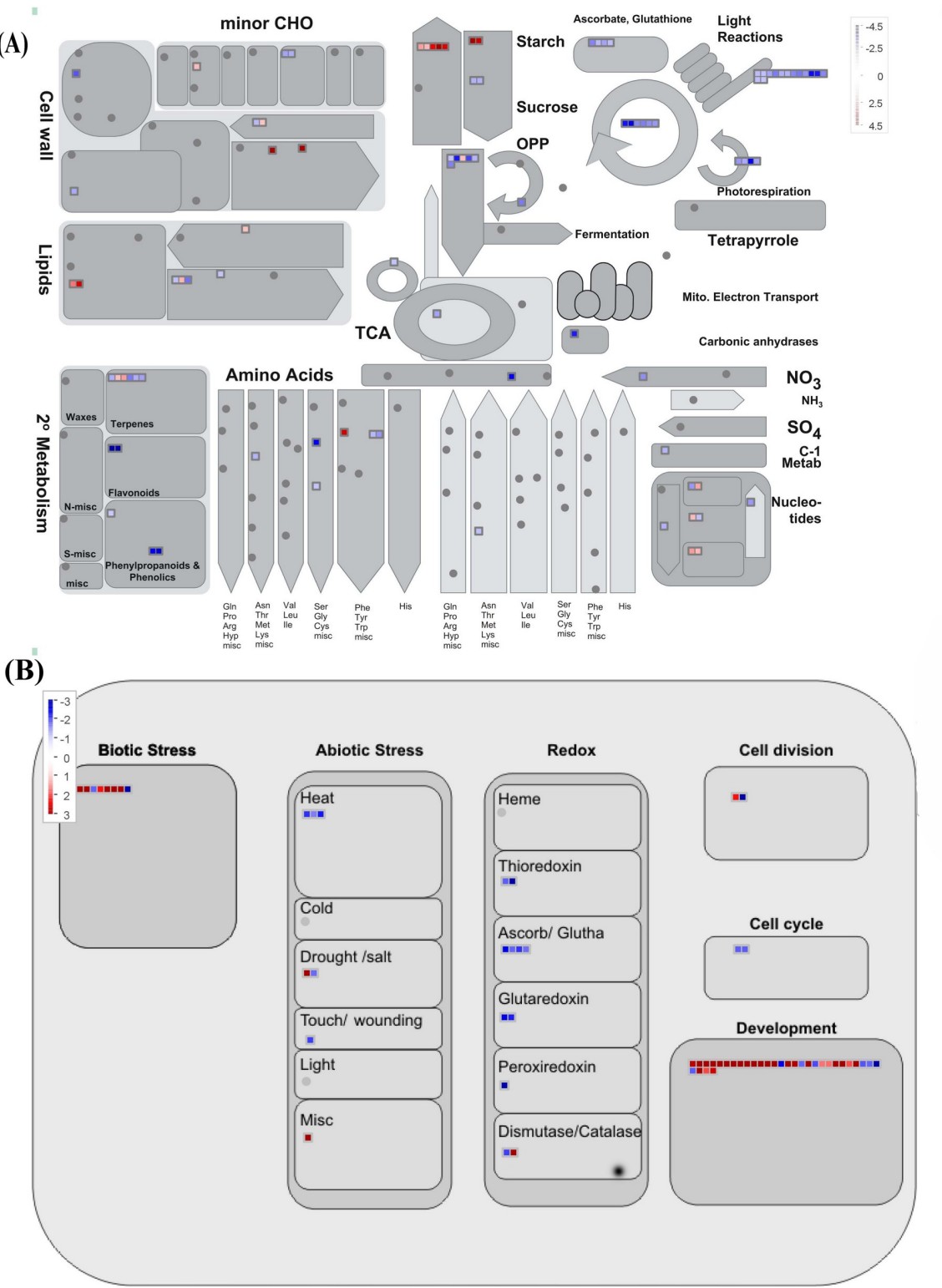

**Fig 5. Functional analysis and metabolism overview of specific DEGs in 33S in response to cold stress.** A. Overview of metbolism. B. Response activities.

nutrient reserve activity to maintain growth under low temperature. This adaptation system seems similar to that of hibernation.

## Conclusions

A comparative RNA-seq analysis was performed to identify low temperature-inducible DEGs in rice seedlings after 7 days of 15°C stress. In total, 938 and 762 DEGs were identified under low-temperature stress in 33S and Nipponbare, respectively. Due to differences in their tolerance to low temperature, these two varieties showed different transcriptional responses. A large number of TF-encoded genes were expressed in response to low-temperature treatment, which indicated that TFs play an important role in cold tolerance. The expression of genes involved in cytokinin and ABA signaling was dramatically induced in response to low-temperature treatment, which suggested that phytohormone played a critical role in the low-temperature response. Furthermore, the expression of most of the DEGs involved in protein processing in the ER pathway was repressed under low-temperature treatment. However, compared with the responses of genes under short-term cold treatment, some of the genes exhibited an opposite response pattern in this paper, indicating that these genes have different functions at different stages of cold treatment. The results of the present study provide a basis for an improved understanding of the molecular mechanism associated with the relatively long-term low-temperature stress response of rice seedlings.

## Supporting information

**S1 Table. Primers used for qRT-PCR in this study.**
(DOC)

**S2 Table. List of genes upregulated in response to low-temperature treatment in 33S.**
(XLS)

**S3 Table. List of genes downregulated in response to low-temperature treatment in 33S.**
(XLS)

**S4 Table. List of genes upregulated in response to low-temperature treatment in Nipponbare.**
(XLS)

**S5 Table. List of genes downregulated in response to low-temperature treatment in Nipponbare.**
(XLS)

**S6 Table. List of 795 DEGs that respond to low temperature only in 33S.**
(XLS)

**S7 Table. GO enrichment analysis of DEGs in 33S.**
(XLS)

**S8 Table. GO enrichment analysis of DEGs in Nipponbare.**
(XLS)

**S9 Table. KEGG pathway enrichment analysis of DEGs in 33S.**
(XLS)

**S10 Table. KEGG pathway enrichment analysis of DEGs in Nipponbare.**
(XLS)

**S11 Table. TF response to low-temperature treatment of 33S.**
(XLS)

**S12 Table. TF response to low-temperature treatment of Nipponbare.**
(XLS)

## Author Contributions

**Conceptualization:** Mingdong Zhu.

**Formal analysis:** Yaying Yu.

**Investigation:** Guohua Tang, Yonghong Duan.

**Project administration:** Yinghong Yu.

**Software:** Xiaoshan Zeng.

**Supervision:** Jianlong Wang.

**Writing – original draft:** Hongjun Xie.

**Writing – review & editing:** Mingdong Zhu.

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
