## [Decision Letter · Decision Letter 0]

23 Jun 2021

PONE-D-21-12476

Comparative transcriptome analysis of the cold resistance of the sterile rice line 33S

PLOS ONE

Dear Dr. Zhu,

Thank you for submitting your manuscript to PLOS ONE. After careful consideration, we feel that it has merit but does not fully meet PLOS ONE’s publication criteria as it currently stands. Therefore, we invite you to submit a revised version of the manuscript that addresses the points raised during the review process.

We look forward to receiving your revised manuscript.

Kind regards,

Jian Zhang

Academic Editor

PLOS ONE

Journal Requirements:

3. We note that Figure 1 in your submission contain copyrighted images. All PLOS content is published under the Creative Commons Attribution License (CC BY 4.0), which means that the manuscript, images, and Supporting Information files will be freely available online, and any third party is permitted to access, download, copy, distribute, and use these materials in any way, even commercially, with proper attribution. For more information, see our copyright guidelines: http://journals.plos.org/plosone/s/licenses-and-copyright.

In the figure caption of the copyrighted figure, please include the following text: “Reprinted from [ref] under a CC BY license, with permission from [name of publisher], original copyright [original copyright year].

Additional Editor Comments (if provided):

please carefully address all the questions raised by both reviewers. In particular, the authors should provide more interpretation of the biological significance behind the RNA seq data. e.g. function-known genes involved in low-temperature response, which has been suggested by reviewer 2. The writing needs a thorough revision, as numerous typos, errors could be found in the text, while some unrelated contents should be excluded to make the logic flow clear.

Reviewers' comments:

Reviewer's Responses to Questions

**Comments to the Author**

1. Is the manuscript technically sound, and do the data support the conclusions?

Reviewer #1: Yes

Reviewer #2: Partly

2. Has the statistical analysis been performed appropriately and rigorously? 

Reviewer #1: Yes

Reviewer #2: I Don't Know

3. Have the authors made all data underlying the findings in their manuscript fully available?

Reviewer #1: No

Reviewer #2: Yes

4. Is the manuscript presented in an intelligible fashion and written in standard English?

Reviewer #1: No

Reviewer #2: No

5. Review Comments to the Author

Reviewer #1: The manuscript drafted by Xie et al. profiled transcriptome changes of cold-resistant rice variety 33S and cold-sensitive variety Nipponbare in response to 7-days’ cold stress at seedling stage. The comparative analyses identified DEGs and regulatory pathways distinguished between 33S and Nipponbare. The study proposed regulation mechanisms in response to cold stress of rice. However, the manuscript hasn’t met the standard for publication at this moment. A major revision is recommended.

Comments:

1. The abstract should be succinct. The very detailed information, such as the ones in parentheses, are suggested to be excluded from the abstract. Grammar issues found in Abstract.

2. I didn’t see any Tables attached in the manuscript. Table 1, 3, and 4 were mentioned in paper (Table 2 is missing) but not shown.

3. Materials and Methods section needs more detailed descriptions. Such as the ‘RNA sequencing (RNA-seq) data analysis’ section: reference genome version needs to be clarified; No QC step or reads trimming? Line 122 needs citation; Line 124, which method used for p-value correction? What’s key parameters used in Tophat and HTseq?

4. Line 92, the seedling for experiment is ‘fifty-one-week-old seedlings’? That does not make sense at all.

5. Figures, figure legends, and Tables are very messy.

1). No figure legend for Figure 1.

2) current Figure 1 legend is actually for Figure 2. For Figure 2 legend, please add a sentence to clarify ‘N’ and ‘CK’. Line 149-153 is a better legend for figure 2 rather than the one in Line 185-186.

3) Supplementary tables should be consistent with their order appear in paper. Line174-179.

4) Figure 3 C and D, what’s Q-vaule? No ‘red line’ in figure 3 as descripted in it's legend.

Figure 3A, what’s the asterisk indicated?

5) Line 300-301, it should be Figure 2 instead of Figure 1.

6) Figure 5 legend is missing.

6. Line 145, inappropriate statement: ‘no 33S seedlings were affected by low temperature’. Since a large number DEGs were identified, there must be affections in 33S. You could say something like ‘no 33s seedlings died after cold treatment’.

7. I didn’t understand the logic in Line 179-182.

8. Throughout the paper, please make sure gene names are in italic format but not for protein names.

9. The RNAseq data should be deposit in public database.

10. Line 170-172, what’s Q <0.05 mean? And also ‘Differences in gene

expression in the six samples were examined using a threshold…’ is not accurate. DEGs are determined by two conditions not individual sample. Line 172-173 is a better description. Combine line 171-173.

Reviewer #2: The authors tried to investigate the low-temperature response mechanisms by comparative transcriptomic analysis between a TGMS variety (33S) and a cold-sensitive variety (Nipponbare). This study is meaningful for our understanding of low-temperature response mechanisms in rice seedling, as well as genetic improvement of cold-tolerant rice in future. However, the manuscript has to be improved in both science and writing for publication in PLOS ONE.

Major comments:

(1) No tables found in this manuscript.

(2) The analysis is insufficient. The authors performed RNA-Seq analysis, to focus on the biological question, they should focus on the specifically biological processes or DEGs in low-temperature response of 33S in contrast with Nipponbare.

(3) To be more convincing, the authors should also analyze the function-known genes involved in low-temperature response, not just common GO and KEGG analysis.

(4) Some descriptions were unrelated to the topic, and the logic is a little bit confusing. For example, the ARR9 is reported to be involved in cold treatment, but it is responsive to low-temperature treatment in the Nipponbare variety, which is a cold-sensitive variety. This cannot explain why 33S is cold-tolerant.

(5) Several conclusions were over speculative from the RNA-seq data throughout the manuscript. For example, the line 250~252 is not supported by only expression of ARRs, which needs more evidences, such as enrichment of GO or KEGG or others. The authors should draw any conclusion throughout the manuscript, cautiously.

Minor comments:

(1) The English writing need to be polished by a native English speaker or language service to correct spelling and grammar errors. For example, ‘Ehe’ should be ‘The’ in line 40. The first word of cold stress in line 48 should be capitalized. The gene names should be italicized in line 135, line 227~229, line 240~247, line 254, line 287~289, line 320~ 337, etc.

(2) Figure legends and figures should be of good shape. Some figure legends were not detailed, such as the means of green and red dots in line 185 (Figure 2). There were no A, B, and C marked in Figure 1. In Figure 1A, the roots showed shorter than control, which is contradictory with the statistical result of Figure 1B. In Figure 1C, ‘33S-Cold vs 33S-Cold’ should be ‘33S-Cold vs 33S-CK’. The statistical test, such as student’s t-test, should be added in Figure 4. Letter numbers have brackets in Figure 5, which is not observed in other Figures.

(3) The section of ‘Illumina RNA-seq and assembly analyses’ is the most basic for RNA-seq analysis, and was uncorrelated to the topic. It should be deleted to make the manuscript more concise or move to “Materials and Methods” section.

6. PLOS authors have the option to publish the peer review history of their article (what does this mean?). If published, this will include your full peer review and any attached files.

Reviewer #1: No

Reviewer #2: No

---

## [Author Response · Author response to Decision Letter 0]

27 Sep 2021

Reply ：We have modified the style of PLoS One's style requirements.

Whilst you may use any professional scientific editing service of your choice, PLOS has partnered with both American Journal Experts (AJE) and Editage to provide discounted services to PLOS authors. Both organizations have experience helping authors meet PLOS guidelines and can provide language editing, translation, manuscript formatting, and figure formatting to ensure your manuscript meets our submission guidelines. To take advantage of our partnership with AJE, visit the AJE website (http://learn.aje.com/plos/) for a 15% discount off AJE services. To take advantage of our partnership with Editage, visit the Editage website (www.editage.com) and enter referral code PLOSEDIT for a 15% discount off Editage services. If the PLOS editorial team finds any language issues in text that either AJE or Editage has edited, the service provider will re-edit the text for free.Upon resubmission, please provide the following:

The name of the colleague or the details of the professional service that edited your manuscript A copy of your manuscript showing your changes by either highlighting them or using track changes (uploaded as a *supporting information* file) . A clean copy of the edited manuscript (uploaded as the new *manuscript* file)”

Reply ：We have completed the revision in AJE.

Reply ：Figure 1 without copyright protection. The pictures and data are obtained by the author in this paper experiment.

Comments:

1. The abstract should be succinct. The very detailed information, such as the ones in parentheses, are suggested to be excluded from the abstract. Grammar issues found in Abs

Reply: we have simplified and modified the syntax of the abstract.

2. I didn’t see any Tables attached in the manuscript. Table 1, 3, and 4 were mentioned in paper (Table 2 is missing) but not shown.

Reply: the form has been placed in the appropriate position in the article.

3. Materials and Methods section needs more detailed descriptions. Such as the ‘RNA sequencing (RNA-seq) data analysis’ section: reference genome version needs to be clarified; No QC step or reads trimming? Line 122 needs citation; Line 124, which method used for p-value correction? What’s key parameters used in Tophat and HTseq?

Reply: Thank you for your suggestions. We have enriched the contents of materials and methods, and added corresponding references.

4. Line 92, the seedling for experiment is ‘fifty-one-week-old seedlings’? That does not make sense at all.

Reply: Sorry, this is a clerical error. We have replaced fifth one week old seeds with fifth one week old seeds.

5. Figures, figure legends, and Tables are very messy.

1). No figure legend for Figure 1.

Reply: the annotation of Figure 1 has been added.

2) current Figure 1 legend is actually for Figure 2. For Figure 2 legend, please add a sentence to clarify ‘N’ and ‘CK’. Line 149-153 is a better legend for figure 2 rather than the one in Line 185-186.

Reply: the description has been added as required and the drawing notes have been modified.

3) Supplementary tables should be consistent with their order appear in paper. Line174-179.

Reply: I'm sorry for our carelessness. Modified as required

4) Figure 3 C and D, what’s Q-vaule? No ‘red line’ in figure 3 as descripted in it's legend.

Figure 3A, what’s the asterisk indicated?

Reply: Q-value has been modified to p-value. The description of the red line has been deleted. The asterisk in Figure 3A has no meaning and has been removed.

5) Line 300-301, it should be Figure 2 instead of Figure 1.

Reply: sorry, it has been modified

6) Figure 5 legend is missing.

Reply: sorry, we have added a drawing note.

6. Line 145, inappropriate statement: ‘no 33S seedlings were affected by low temperature’. Since a large number DEGs were identified, there must be affections in 33S. You could say something like ‘no 33s seedlings died after cold treatment’.

Reply: Thank you for your suggestion. We have modified this part according to your suggestion

7. I didn’t understand the logic in Line 179-182.

Reply: Thank you for your suggestion. We have recounted this part.

8. Throughout the paper, please make sure gene names are in italic format but not for protein names.

Reply: we have checked and modified the format in the full text as required.

9. The RNAseq data should be deposit in public database.

Reply: we have submitted data in the public database and provided data acquisition channels in the paper.

10. Line 170-172, what’s Q <0.05 mean? And also ‘Differences in gene

expression in the six samples were examined using a threshold…’ is not accurate. DEGs are determined by two conditions not individual sample. Line 172-173 is a better description. Combine line 171-173.

Reply: Thank you for your comments. Q < 0.05 has been revised to P < 0.05. We re described the contents of lines 170-172.

Reviewer #2: The authors tried to investigate the low-temperature response mechanisms by comparative transcriptomic analysis between a TGMS variety (33S) and a cold-sensitive variety (Nipponbare). This study is meaningful for our understanding of low-temperature response mechanisms in rice seedling, as well as genetic improvement of cold-tolerant rice in future. However, the manuscript has to be improved in both science and writing for publication in PLOS ONE.

Major comments:

(1) No tables found in this manuscript.

Reply: we have put the picture in the right place according to the text.

(2) The analysis is insufficient. The authors performed RNA-Seq analysis, to focus on the biological question, they should focus on the specifically biological processes or DEGs in low-temperature response of 33S in contrast with Nipponbare.

Reply: Thank you for your suggestion. We added the analysis of differential genes in the manuscript.

(3) To be more convincing, the authors should also analyze the function-known genes involved in low-temperature response, not just common GO and KEGG analysis.

Reply: Thank you for your suggestion. We analyzed and discussed the known genes involved in hypothermia (such as) in the revised manuscript.

(4) Some descriptions were unrelated to the topic, and the logic is a little bit confusing. For example, the ARR9 is reported to be involved in cold treatment, but it is responsive to low-temperature treatment in the Nipponbare variety, which is a cold-sensitive variety. This cannot explain why 33S is cold-tolerant.

Reply: Thank you for your suggestion. However, the current research results show that the mechanism of plant low temperature tolerance is very complex, and many genes are involved. Although 33S material has stronger adaptability to low temperature than Nipponbare, there must be some common genes responding to low temperature stress at the same time.

(5) Several conclusions were over speculative from the RNA-seq data throughout the manuscript. For example, the line 250~252 is not supported by only expression of ARRs, which needs more evidences, such as enrichment of GO or KEGG or others. The authors should draw any conclusion throughout the manuscript, cautiously.

Reply: Thank you for your comments. The result data provided in our paper is indeed insufficient. We have modified the description of the conclusion。

Minor comments:

(1) The English writing need to be polished by a native English speaker or language service to correct spelling and grammar errors. For example, ‘Ehe’ should be ‘The’ in line 40. The first word of cold stress in line 48 should be capitalized. The gene names should be italicized in line 135, line 227~229, line 240~247, line 254, line 287~289, line 320~ 337, etc.

Reply: We apologize for our carelessness. We have carefully checked and revised the full manuscript.

(2) Figure legends and figures should be of good shape. Some figure legends were not detailed, such as the means of green and red dots in line 185 (Figure 2). There were no A, B, and C marked in Figure 1. In Figure 1A, the roots showed shorter than control, which is contradictory with the statistical result of Figure 1B. In Figure 1C, ‘33S-Cold vs 33S-Cold’ should be ‘33S-Cold vs 33S-CK’. The statistical test, such as student’s t-test, should be added in Figure 4. Letter numbers have brackets in Figure 5, which is not observed in other Figures.

Reply: Thank you for your comments. We have modified the picture format and annotation according to your comments.

(3) The section of ‘Illumina RNA-seq and assembly analyses’ is the most basic for RNA-seq analysis, and was uncorrelated to the topic. It should be deleted to make the manuscript more concise or move to “Materials and Methods” section.

Reply: Thank you for your suggestion. We have removed this part.

---

## [Decision Letter · Decision Letter 1]

1 Nov 2021

PONE-D-21-12476R1Comparative transcriptome analysis of the cold resistance of the sterile rice line 33SPLOS ONE

Dear Dr. Zhu,

Thank you for submitting your manuscript to PLOS ONE. After careful consideration, we feel that it has merit but does not fully meet PLOS ONE’s publication criteria as it currently stands. Therefore, we invite you to submit a revised version of the manuscript that addresses the points raised during the review process.

We look forward to receiving your revised manuscript.

Kind regards,

Jian Zhang

Academic Editor

PLOS ONE

Journal Requirements:

Additional Editor Comments:

As you may find from attached, reviewer 2 had no more comments, while reviewer 1 is still not satisfied with some of the data interpretation and requested further revision. I believe this paper will be accepted upon careful revisions.

Reviewers' comments:

Reviewer's Responses to Questions

**Comments to the Author**

1. If the authors have adequately addressed your comments raised in a previous round of review and you feel that this manuscript is now acceptable for publication, you may indicate that here to bypass the “Comments to the Author” section, enter your conflict of interest statement in the “Confidential to Editor” section, and submit your "Accept" recommendation.

Reviewer #1: (No Response)

Reviewer #2: All comments have been addressed

2. Is the manuscript technically sound, and do the data support the conclusions?

Reviewer #1: Partly

Reviewer #2: Yes

3. Has the statistical analysis been performed appropriately and rigorously? 

Reviewer #1: No

Reviewer #2: N/A

4. Have the authors made all data underlying the findings in their manuscript fully available?

Reviewer #1: Yes

Reviewer #2: Yes

5. Is the manuscript presented in an intelligible fashion and written in standard English?

Reviewer #1: Yes

Reviewer #2: Yes

6. Review Comments to the Author

Reviewer #1: The revised version manuscript entitled with ‘Comparative transcriptome analysis of the cold resistance of the sterile rice line 33S’ was improved and basically addressed most of my concerns. However, a couple of points need to be clarified before acceptation. Especially, please specify criterion used for GO and KEGG enrichment analyses as some key conclusions were drawn from them (Point 2.2 in below).

1. In the abstract section, it uncommon to describe pathway by ‘reduced’ or ‘increased’. It can be replaced by ‘repressed’ or ‘enhanced’.

2. Materials and Methods section:

1) please specify the reference genome version, such as v7 or something else, used for RNAseq analysis.

2). There’s no criterion described for GO and KEGG pathway enrichment analyses. The E value <= 10-5 is expect value during BLAST but not for determining statistical significance for enrichment. I checked Table S8, S9, and S10 as the results for functional enrichment analyses and it seems most terms with p-value >0.05 which indicated statistically unsignificant. Since some conclusions were based on the functional enrichment analyses, please make it clear.

3. Figure 1: the labels for ‘33S’ and ‘Nipponbare’ are wrong; no letters indicated significant difference between comparison, but mentioned in legend; The error bar in Figure1C looked inconsistent with Figure 1B, did you used standard error or standard deviation for error bar?

4. Above Figure 2, on page 7, it should be ‘In contrast, more up-regulated genes were identified in Nipponbare variety’ instead of ‘down-regulated’.

5. Figure 2 legend, ‘|log2FoldChange| >1’ is actually ‘|log2FoldChange| >=1’ to be consistant with Materials and Methods?

6. Figure 3C and D, does the ‘qvalue’ in figure actually mean ‘pvalue’? What’s the ‘rich factor’ indicated?

Reviewer #2: (No Response)

7. PLOS authors have the option to publish the peer review history of their article (what does this mean?). If published, this will include your full peer review and any attached files.

Reviewer #1: No

Reviewer #2: No

---

## [Author Response · Author response to Decision Letter 1]

6 Dec 2021

6. Review Comments to the Author

Reviewer #1: The revised version manuscript entitled with ‘Comparative transcriptome analysis of the cold resistance of the sterile rice line 33S’ was improved and basically addressed most of my concerns. However, a couple of points need to be clarified before acceptation. Especially, please specify criterion used for GO and KEGG enrichment analyses as some key conclusions were drawn from them (Point 2.2 in below).

1. In the abstract section, it uncommon to describe pathway by ‘reduced’ or ‘increased’. It can be replaced by ‘repressed’ or ‘enhanced’.

Response: Thanks for your suggestion. They were changed (line 31 and line 33, page 2).

2. Materials and Methods section:

1) please specify the reference genome version, such as v7 or something else, used for RNAseq analysis.

Response: the genome we used was GCF_000005425.2_Build_4.0_genomic, and we added it in the M&M.

2). There’s no criterion described for GO and KEGG pathway enrichment analyses. The E value <= 10-5 is expect value during BLAST but not for determining statistical significance for enrichment. I checked Table S8, S9, and S10 as the results for functional enrichment analyses and it seems most terms with p-value >0.05 which indicated statistically unsignificant. Since some conclusions were based on the functional enrichment analyses, please make it clear.

Response: Thanks for your suggestion, we had corrected these descriptions, only that p<0.05 were described as significant, and those p>0.05 were considered not to enriched. But we still thought the DEGs in such pathways were valuable to reveal the regulation mechanism of cold tolerance. Please see the line 157-164, page 7, the corrections were made. 

3. Figure 1: the labels for ‘33S’ and ‘Nipponbare’ are wrong; no letters indicated significant difference between comparison, but mentioned in legend; The error bar in Figure1C looked inconsistent with Figure 1B, did you used standard error or standard deviation for error bar?

Response: we both used the standard deviation (SD), and this description was added into figure legends. Actually, the error bar in Figure1C looked inconsistent with Figure 1B, perhaps the root length were affected by cold leading to bigger difference. 

4. Above Figure 2, on page 7, it should be ‘In contrast, more up-regulated genes were identified in Nipponbare variety’ instead of ‘down-regulated’.

Response: Thanks, it has been changed. 

5. Figure 2 legend, ‘|log2FoldChange| >1’ is actually ‘|log2FoldChange| >=1’ to be consistant with Materials and Methods?

Response: Thanks, it has been changed. 

6. Figure 3C and D, does the ‘qvalue’ in figure actually mean ‘pvalue’? What’s the ‘rich factor’ indicated? 

Response: yes, qvalue is a adjust pvalue. And the rich factor means the ratio of DEGs in total specific annotated pathway/term. 

Reviewer #2: (No Response)

---

## [Decision Letter · Decision Letter 2]

13 Dec 2021

Comparative transcriptome analysis of the cold resistance of the sterile rice line 33S

PONE-D-21-12476R2

Dear Dr. Zhu,

We’re pleased to inform you that your manuscript has been judged scientifically suitable for publication and will be formally accepted for publication once it meets all outstanding technical requirements.

Kind regards,

Jian Zhang

Academic Editor

PLOS ONE

Additional Editor Comments (optional):

Reviewers' comments:

Reviewer's Responses to Questions

**Comments to the Author**

1. If the authors have adequately addressed your comments raised in a previous round of review and you feel that this manuscript is now acceptable for publication, you may indicate that here to bypass the “Comments to the Author” section, enter your conflict of interest statement in the “Confidential to Editor” section, and submit your "Accept" recommendation.

Reviewer #1: All comments have been addressed

2. Is the manuscript technically sound, and do the data support the conclusions?

Reviewer #1: Yes

3. Has the statistical analysis been performed appropriately and rigorously? 

Reviewer #1: Yes

4. Have the authors made all data underlying the findings in their manuscript fully available?

Reviewer #1: Yes

5. Is the manuscript presented in an intelligible fashion and written in standard English?

Reviewer #1: Yes

6. Review Comments to the Author

Reviewer #1: (No Response)

7. PLOS authors have the option to publish the peer review history of their article (what does this mean?). If published, this will include your full peer review and any attached files.

Reviewer #1: No

---

## [Editor Report · Acceptance letter]

6 Jan 2022

PONE-D-21-12476R2 

Comparative transcriptome analysis of the cold resistance of the sterile rice line 33S 

Dear Dr. Wang:

I'm pleased to inform you that your manuscript has been deemed suitable for publication in PLOS ONE. Congratulations! Your manuscript is now with our production department. 

Kind regards, 

on behalf of

Professor Jian Zhang 

Academic Editor

PLOS ONE